# Model misspecification, measurement error, and apparent supralinearity in the concentration-response relationship between PM$_{2.5}$ and mortality

**Garrett Glasgow** [1]*, **Bharat Ramkrishnan** [2], **Anne E. Smith** [2]

**1** NERA Economic Consulting, San Francisco, California, United States of America, **2** NERA Economic Consulting, Washington, District of Columbia, United States of America

☯ These authors contributed equally to this work.

\* garrett.glasgow@nera.com

**Data Availability Statement:** The simulated cohorts underlying the results presented in the study are available from the Harvard dataverse:

## Abstract

A growing number of studies have produced results that suggest the shape of the concentration-response (C-R) relationship between PM$_{2.5}$ exposure and mortality is "supralinear" such that incremental risk is higher at the lowest exposure levels than at the highest exposure levels. If the C-R function is in fact supralinear, then there may be significant health benefits associated with reductions in PM$_{2.5}$ below the current US National Ambient Air Quality Standards (NAAQS), as each incremental tightening of the PM$_{2.5}$ NAAQS would be expected to produce ever-greater reductions in mortality risk. In this paper we undertake a series of tests with simulated cohort data to examine whether there are alternative explanations for apparent supralinearity in PM$_{2.5}$ C-R functions. Our results show that a linear C-R function for PM$_{2.5}$ can falsely appear to be supralinear in a statistical estimation process for a variety of reasons, such as spatial variation in the composition of total PM$_{2.5}$ mass, the presence of confounders that are correlated with PM$_{2.5}$ exposure, and some types of measurement error in estimates of PM$_{2.5}$ exposure. To the best of our knowledge, this is the first simulation-based study to examine alternative explanations for apparent supralinearity in C-R functions.

## 1. Introduction

A major driver of policy decisions regarding the level at which to set the US National Ambient Air Quality Standards (NAAQS) for particulate matter is evidence of increased mortality risk due to long term exposure to PM$_{2.5}$ (particulate matter with a diameter of 2.5 micrometers or less). Numerous epidemiological studies over the past several decades have found a statistical association between PM$_{2.5}$ and mortality, with the strongest effects usually reported from chronic exposure studies that compare survival outcomes of cohorts in different communities with differing ambient concentrations of PM$_{2.5}$ [1–7].

Nearly all of the major cohort epidemiological analyses undertaken to date have assumed a linear slope for the concentration-response (C-R) function that determines the association

https://dataverse.harvard.edu/dataset.xhtml?persistentId=doi:10.7910/DVN/JTBYUS.

**Funding:** This work was funded by the Electric Power Research Institute (EPRI). URL: https://www.epri.com The funders had no role in study design, data collection and analysis, decision to publish, or preparation of the manuscript.

**Competing interests:** The authors have declared that no competing interests exist.

between exposure to PM$_{2.5}$ and mortality. Under this assumption, a given improvement in ambient air quality (*e.g.*, a reduction of 1 μg/m$^3$ in the ambient concentration of PM$_{2.5}$) is expected to produce the same reduction in mortality risk, regardless of the initial concentration of PM$_{2.5}$.

Relaxing this assumption of linearity and determining the true "shape" of the C-R function is a fundamental challenge in determining whether and to what extent tightening air quality standards for PM$_{2.5}$ will result in public health benefits, particularly at relatively low PM$_{2.5}$ concentrations. For example, if the C-R function is "sublinear," such that incremental risk is lower at the lowest exposure levels than at the highest exposure levels, then there will be diminishing returns to each incremental tightening of air quality standards. Conversely, if the C-R function is "supralinear," such that incremental risk is higher at the lowest exposure levels than at the highest exposure levels, then there may be significant health benefits associated with reductions in PM$_{2.5}$ below the current NAAQS [8], as each incremental tightening of air quality standards would be expected to produce increasingly larger reductions in mortality risk [9].

A growing number of studies have produced results that suggest the shape of the C-R relationship between PM$_{2.5}$ exposure and mortality risk is supralinear [3,8,10–14]. If true, this would have important implications for setting the NAAQS for PM$_{2.5}$ and the process by which NAAQS levels have traditionally been justified.

However, there are a number of factors that could affect the relationship estimated between PM$_{2.5}$ exposure and mortality risk, indicating an apparently supralinear C-R function, even if the true relationship between PM$_{2.5}$ exposure and mortality is linear. In this study we undertake a series of simulations to test whether the C-R relationship between PM$_{2.5}$ exposure and mortality risk could plausibly be linear, with the apparent supralinearity estimated by some studies actually the result of biases caused by other factors associated with the epidemiological data. Simulation is an important supplement to standard epidemiological investigations that rely on observed data. In standard epidemiology studies with observed cohort data, the true shape of the C-R relationship is unknown to epidemiological researchers, so it is not possible to determine how factors associated with PM$_{2.5}$ might be biasing the estimated shape of the C-R function. Our simulations allow us to test the plausibility of various alternative explanations for apparent supralinearity in a way that is not possible with real-world data. To the best of our knowledge, this is the first simulation-based study to examine alternative explanations for apparent supralinearity in C-R functions.

We consider three alternative explanations for apparent supralinearity in a C-R function. The first is whether spatial variation in the composition of total PM$_{2.5}$ mass can cause the false appearance of supralinearity in statistical estimates. Under this scenario, total PM$_{2.5}$ mass is composed of both toxic and non-toxic constituents, with the fraction of the mass that is toxic affecting mortality through a linear C-R function. Even though the true C-R function for toxic PM$_{2.5}$ is linear, if the fraction of the mass that is toxic is higher in areas with lower total PM$_{2.5}$ mass, the estimated C-R function based on total PM$_{2.5}$ mass may appear to be supralinear.

The second alternative explanation we test is whether apparent supralinearity can be arise in cases where PM$_{2.5}$ exposure has been serving as a proxy for some other risk factor. Under this scenario, if the alternative risk factor is disproportionately large in areas with higher PM$_{2.5}$ concentrations, the estimated C-R function for PM$_{2.5}$ may appear to be supralinear, even if the true C-R function for the alternative risk factor is linear.

The third alternative explanation we test examines whether exposure misclassification (classical measurement error) related to measured PM$_{2.5}$ concentrations can result in apparent supralinearity in statistical estimates. We examine two patterns of measurement error, one under which exposure misclassification is disproportionately large in areas with higher PM$_{2.5}$

concentrations, and one under which exposure misclassification is disproportionately small in areas with higher PM$_{2.5}$ concentrations. Previous studies have demonstrated that measurement error leads to attenuated estimates of the slope of a C-R function, which in turn can affect the estimated shape of the C-R function [15,16]. If the measurement error varies across the range of measured PM$_{2.5}$ concentrations, this could attenuate some portions of the C-R function more than others, leading to apparent supralinearity in the estimation of the C-R function.

## 2. Methods

### 2.1. Generating the simulated cohorts

For each alternative explanation for apparent supralinearity we tested, we generated a data set of simulated cohorts. Each simulated cohort consisted of populations from 100 different hypothetical cities, each with 10,000 simulated individuals, for a total cohort size of 1 million individuals. Each simulated cohort was tracked over time for 20 years beginning in the year 2000. The baseline mortality rate for the simulated individuals in our cohort was calculated using cohort life tables compiled by the US Social Security Administration [17]. These life tables give the probability of mortality at each age based on birth year and sex, with birth year reported in 10-year increments from 1900 to 2100. We used linear interpolation to assign mortality probabilities for birth years that fell between the years for which data are reported in the life table. Note that baseline mortality in our simulations is for all-cause mortality, and not just mortality due to causes that may be influenced by PM$_{2.5}$ exposure. Replacing all-cause mortality with a more limited measure of mortality would reduce the baseline mortality in our simulations, but would not affect the simulation results, which measure the relative risk due to increased (simulated) PM$_{2.5}$ exposure, and thus are not dependent on the baseline starting point. However, the effect of PM$_{2.5}$ exposure on absolute risk cannot be inferred from our simulations.

For each simulated individual, the birth year was calculated by subtracting the age assigned to that individual from the first calendar year of the simulation (2000), while the probability of dying for the simulated individual in each year of the simulation was assigned based on the mortality probabilities calculated above.

To eliminate variation in the baseline mortality rate based on age and sex that would otherwise make the detection of the shape of the C-R function more difficult, we limited the cohort we generated to consist only of males aged 60 at the start of the simulation. The relative risk of PM$_{2.5}$ exposure based on sex cannot be inferred from our simulations, especially since the effect of PM$_{2.5}$ exposure on mortality may differ by sex [18]. Variation in age and sex could also be introduced into the cohort and is an area for future study.

For each simulated individual in each year of the simulation, mortality outcome was determined by a random draw from a Bernoulli distribution with the probability of mortality given by $P_{ijt}$, which is described in Section 2.3 below. Even though all simulated individuals in the cohorts have the same baseline mortality, actual mortality outcomes will vary due to variation in the random draws across cohorts even without the influence of other risk factors.

To mitigate against the effects of this random variation in mortality, in most cases we generated 10 simulated cohorts for each specific scenario we examined. As explained below, we tested 30 different scenarios related to spatial variation in the composition of total PM$_{2.5}$ mass (and so generate 300 different cohorts in this case), and 44 different scenarios related to an alternative risk factor for which PM$_{2.5}$ serves as a proxy (and so generate 440 different cohorts in this case). The only exceptions to this are the simulations examining the effect of measurement error–in that case, we generated four simulated cohorts (one for each of the 4 hazard ratios we examine, as explained below).

## 2.2. The risk factors in each set of simulations

The definition of the risk factor varies across the alternative explanations that we tested. A dataset with PM$_{2.5}$ concentrations for the 100 hypothetical cities that are generally consistent with those in the U.S. was created using a triangular distribution function, with the minimum, maximum and mode of the distribution specified to be 5, 20 and 10 μg/m$^3$, respectively. We assumed no variation in PM$_{2.5}$ concentrations over time. Variation in PM$_{2.5}$ concentrations over time could also be examined in the simulation and is an area for future study. For the simulations examining measurement error, these PM$_{2.5}$ concentrations are the risk factor.

For the simulations examining spatial variation in the composition of total PM$_{2.5}$ mass, the risk factor is the fraction of total PM$_{2.5}$ mass that is toxic. We examined 30 different scenarios, with the toxic fraction of PM$_{2.5}$ increasing with total PM$_{2.5}$ concentrations in some cases and decreasing in others. For each scenario, we selected the fraction of toxic PM$_{2.5}$ for the cities with the highest and lowest total PM$_{2.5}$ concentrations and determined the fraction of toxic PM$_{2.5}$ for the remaining cities through linear interpolation. Note that these simulations do not critically rely on our simplifying assumption that some types of PM$_{2.5}$ are non-toxic (even natural sources of PM$_{2.5}$ such as desert dust are likely have some degree of toxicity [19]). Rather, it is the relative toxicity of the PM$_{2.5}$ concentrations that matters for our simulation results.

For the simulations examining some other risk factor for which PM$_{2.5}$ exposure serves as a proxy, we examined 44 different scenarios, with the amount of exposure to the risk factor varying with PM$_{2.5}$ concentrations. As with the scenarios examining the toxic fraction of PM$_{2.5}$, we first selected exposures to the risk factor based on some multiple of the PM$_{2.5}$ concentration for the cities with the lowest and highest PM$_{2.5}$ concentrations and determined exposure to the risk factor for the remaining cities through linear interpolation. Note that unlike in the toxic fraction scenarios, the risk factor here is not a subset of total PM$_{2.5}$ mass, so the values of the risk factor can exceed the PM$_{2.5}$ concentration in some or all hypothetical cities.

## 2.3. The concentration-response functions

In each hypothetical city in each year, a risk factor was assumed to influence the probability of mortality. In all sets of simulations, the entire risk from exposure to a risk factor occurs in the same year as exposure (*i.e.*, there are no lags or cumulative effects). For all alternative explanations we tested, we used C-R functions that were linear on the log hazard scale. Specifically, the probability of mortality ($P_{ijt}$) for simulated individual $i$ in city $j$ at time $t$ was calculated as:

$$P_{ijt} = B_{it} \times h^{PM2.5(j)}$$

where $B_{it}$ is the baseline probability of mortality for simulated individual $i$ at time $t$, *PM2.5(j)* is the PM$_{2.5}$ exposure in city $j$, and $h$ is the hazard ratio related to exposure to the risk factor.

For the simulations examining the composition of total PM$_{2.5}$ mass, the hazard ratio of the toxic fraction of PM$_{2.5}$ was set to 1.05 per 1 μg/m$^3$. The hazard ratio of the risk factor for which PM$_{2.5}$ serves as a proxy was set to 1.05 per unit. For the simulations testing the effect of measurement error, we consider hazard ratios for PM$_{2.5}$ of 1.005, 1.01, 1.02, and 1.03 per 1 μg/m$^3$.

## 2.4. Measurement error

For all alternative explanations for apparent supralinearity except for those considering the effect of measurement error, our statistical models measure the association between mortality and total PM$_{2.5}$ concentrations. For the measurement error simulations, our statistical models measure the association between mortality and "observed" PM$_{2.5}$ concentrations, which are the total PM$_{2.5}$ concentrations adjusted by random draws to simulate measurement error. This

is a type of "classical" measurement error, in which the measured level of ambient pollution differs from the true value for some reason (*e.g.*, imprecision in the monitoring instrument) [16,20].

We considered two different patterns of measurement error. The first pattern was one under which exposure misclassification is disproportionately large in areas with higher PM$_{2.5}$ concentrations (a positive correlation between measurement error and PM$_{2.5}$ concentrations). Each hypothetical city was assigned an observed PM$_{2.5}$ value drawn from a normal distribution with a mean equal to the true PM$_{2.5}$ concentration and a standard deviation equal to a percentage of the true PM$_{2.5}$ concentration. We considered cases in which the standard deviation was equal to 5, 10, and 20 percent of the true PM$_{2.5}$ concentration (low, medium, and high measurement error). To avoid excessively large absolute errors, error draws that exceeded ± 5 μg/m$^3$ were discarded and resampled.

The second pattern was one under which exposure misclassification is disproportionately small in areas with higher PM$_{2.5}$ concentrations (a negative correlation between measurement error and PM$_{2.5}$ concentrations). As with the positive correlation case, the observed PM$_{2.5}$ values were drawn from a normal distribution truncated at ± 5 μg/m$^3$, with a mean equal to the true PM$_{2.5}$ concentration and a standard deviation equal to a percentage of the true PM$_{2.5}$ concentration. The standard deviations for the negative correlation cases were approximately the inverse of those in the positive correlation cases and were created by dividing the true PM$_{2.5}$ concentration by 15, 20, or 23 (high, medium, and low measurement error).

For both the positive and negative correlation cases, we also considered draws of the observed PM$_{2.5}$ values from normal distributions truncated at ± 4 μg/m$^3$ and ± 6 μg/m$^3$.

For each of the 6 patterns of measurement error (three positive correlations and three negative correlations), 100 separate sets of city-specific observed PM$_{2.5}$ values were created for each of the four cohorts (for hazard ratios of 1.005, 1.01, 1.02, and 1.03). The middle 95 percent of the measurement error distributions in the positive and negative correlation cases are presented in Figs 1 and 2.

**Simulations examining true supralinearity.**   In addition to the three sets of simulations examining alternative explanations for apparent supralinearity described above, we also conducted a set of simulations examining scenarios in which the C-R function was in fact supralinear in order to verify that our methods for detecting supralinearity were effective.

For this set of simulations, the log hazard for each supralinear C-R function was derived from the cumulative density function (CDF) of the logistic distribution. The argument for the logistic CDF was the difference between the PM$_{2.5}$ concentration in a hypothetical city and the minimum PM$_{2.5}$ concentration. Changing the values for the scale parameter of the logistic distribution changes the shape of the distribution, producing different degrees of supralinearity. The probability of mortality ($P_{ijt}$) for simulated individual $i$ in city $j$ at time $t$ was calculated as in Section 2.3. The four C-R functions we examined in our simulations (three supralinear plus a linear function) are presented in Fig 3.

In each of these C-R functions, the PM$_{2.5}$ concentrations are the true risk factor. We generated 10 simulated cohorts for each of the 3 degrees of supralinearity plus 10 simulated cohorts for the linear case (for a total of 40 simulated cohorts).

## 2.5. Assessing apparent supralinearity

There is no formal definition of "supralinearity" in the literature, beyond a C-R function under which incremental risk is higher at the lowest exposure levels than at the highest exposure levels. Fig 4 presents a selection of the C-R functions generated by our simulations drawn from the PM$_{2.5}$ composition simulations (the label on each subfigure indicates the toxic

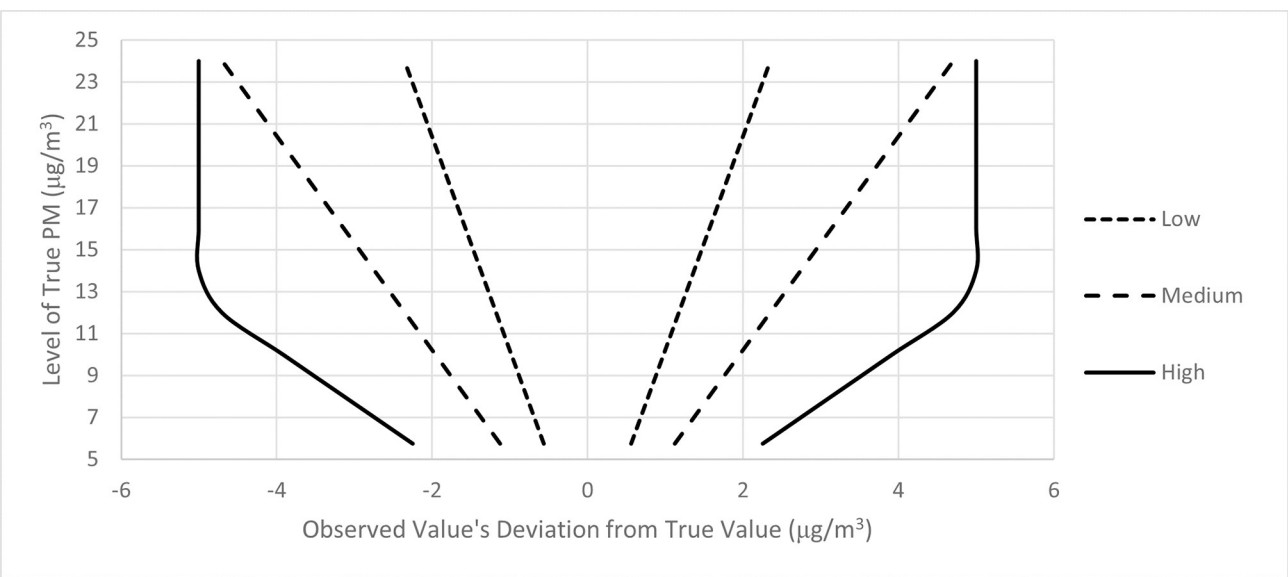

**Fig 1. Middle 95% of measurement error distribution (truncated at ± 5 μg/m³) as function of true PM₂.₅, positive correlation.**

fraction of PM$_{2.5}$ for the lowest and highest total PM$_{2.5}$ concentrations). The figures in the third and fourth rows of Fig 4 would generally be regarded as supralinear based on visual inspection, but there is no simple rule of thumb (such as concavity) that allows for a simple statistical test. Further research is needed to develop a formal definition and statistical test for supralinearity.

For each simulated cohort, we assessed indication of supralinearity by first estimating a Cox proportional hazards (PH) model using the measures of PM exposure described above, with the hazard ratio associated with PM$_{2.5}$ estimated using a natural cubic spline with 2 degrees of freedom. Previous studies have used splines in Cox PH models to assess supralinearity [10,12,14].

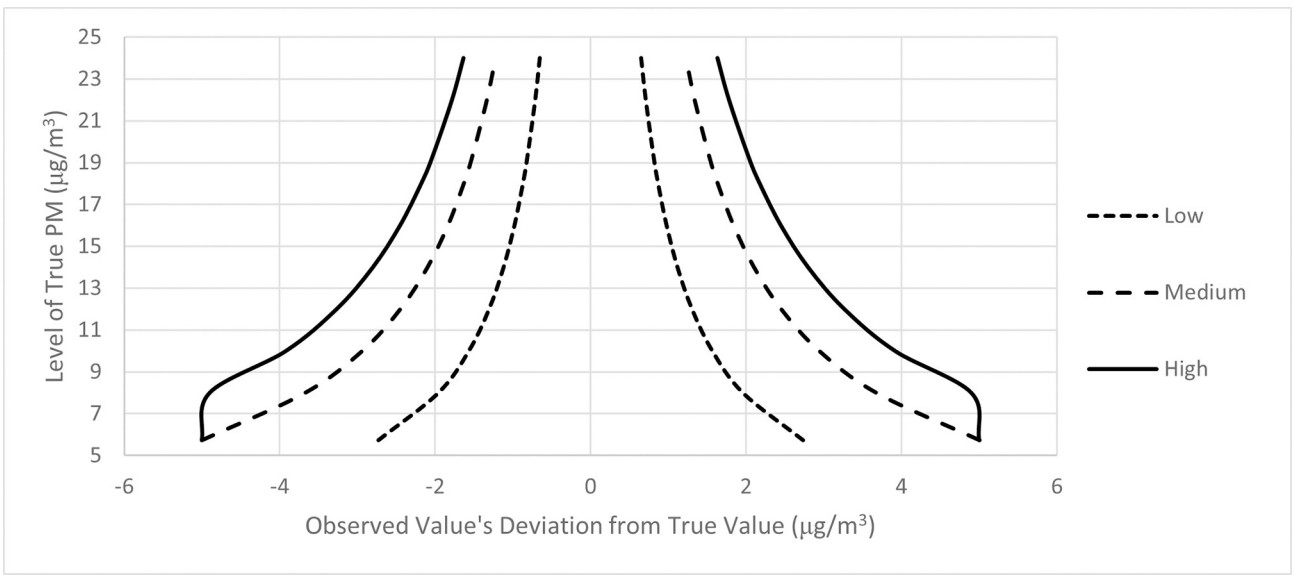

**Fig 2. Middle 95% of measurement error distribution (truncated at ± 5 μg/m³) as function of true PM₂.₅, negative correlation.**

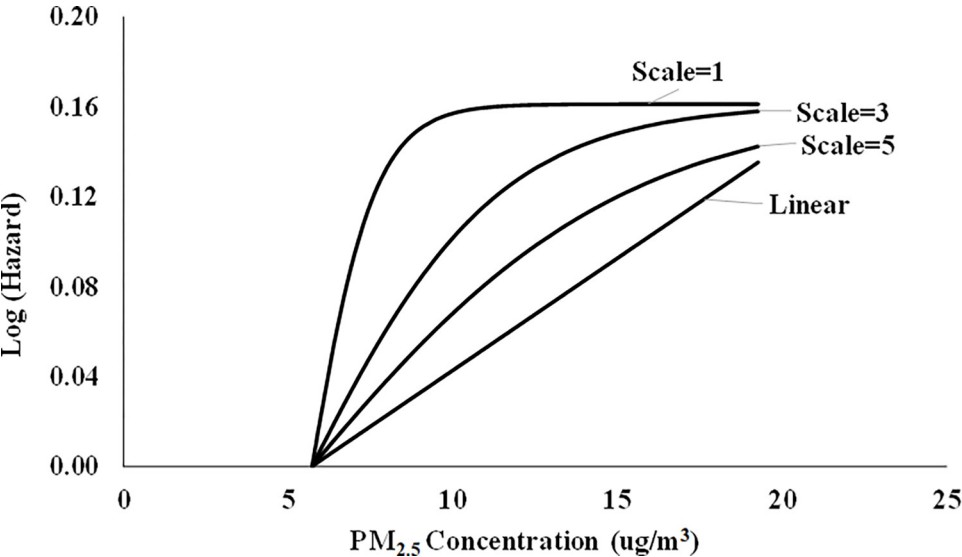

**Fig 3. Supralinear C-R functions.**

Several previous studies have assessed supralinearity through a simple inspection of a plot of the estimated hazard ratio from the Cox PH model across the range of $PM_{2.5}$. However, this method is subjective, and impractical to apply to the large number of simulations we were to undertake. Thus, we developed a test for supralinearity based on a comparison of the slope of the estimated hazard ratio over the first (lower) half of the $PM_{2.5}$ values used in the Cox PH model (Slope A) to the slope of the estimated hazard ratio over the second (higher) half of the $PM_{2.5}$ values (Slope B). The slopes are calculated based on the hazard ratios for the lowest and highest values of $PM_{2.5}$ that correspond to each half (i.e., the lower and higher half) of the $PM_{2.5}$ values. An example of the two slopes calculated from the estimated hazard ratio is presented in Fig 5.

If the assessed C-R function is linear, the ratio of Slope A to Slope B will be equal to one. For a supralinear C-R function, Slope A will be greater than Slope B, leading to a slope ratio greater than one. Conversely, for a sublinear C-R function, Slope A will be less than Slope B, leading to a slope ratio less than one. In our tests, we categorize a C-R function as supralinear if the estimate of the slope ratio is greater than 1.2, sublinear if the estimate of the slope ratio is less than 0.8, and linear is the estimated slope ratio falls between 0.8 and 1.2.

Note that in the set of simulations examining the effect of measurement error, the estimated "true" C-R functions for the cohorts before applying measurement error were not perfectly linear due to random variation in mortality across cities, even though a linear C-R function was used to generate the data. This was particularly so for the case for the lowest hazard ratio for $PM_{2.5}$ we considered (1.005), where one would expect random variation to play the largest role. An application of the relative slope ratio test to the linear C-R function calculated using a hazard ratio of 1.005 indicated apparent sublinearity (with a relative slope ratio of 0.74). The remaining cohorts were correctly estimated to have linear C-R functions (with relative slope ratios of 0.97, 0.97, and 1.03 for hazard ratios 1.01, 1.02, and 1.03, respectively). Thus, to control for deviations from linearity in the shape of the C-R function associated with no measurement error, and to ensure we could detect shifts in the supralinear direction due to measurement error, the slope ratio test for these simulations was applied relative to the C-R function with no measurement error (*i.e.*, we test whether measurement error shifts the apparent C-R function in a supralinear direction relative to the "true" C-R function).

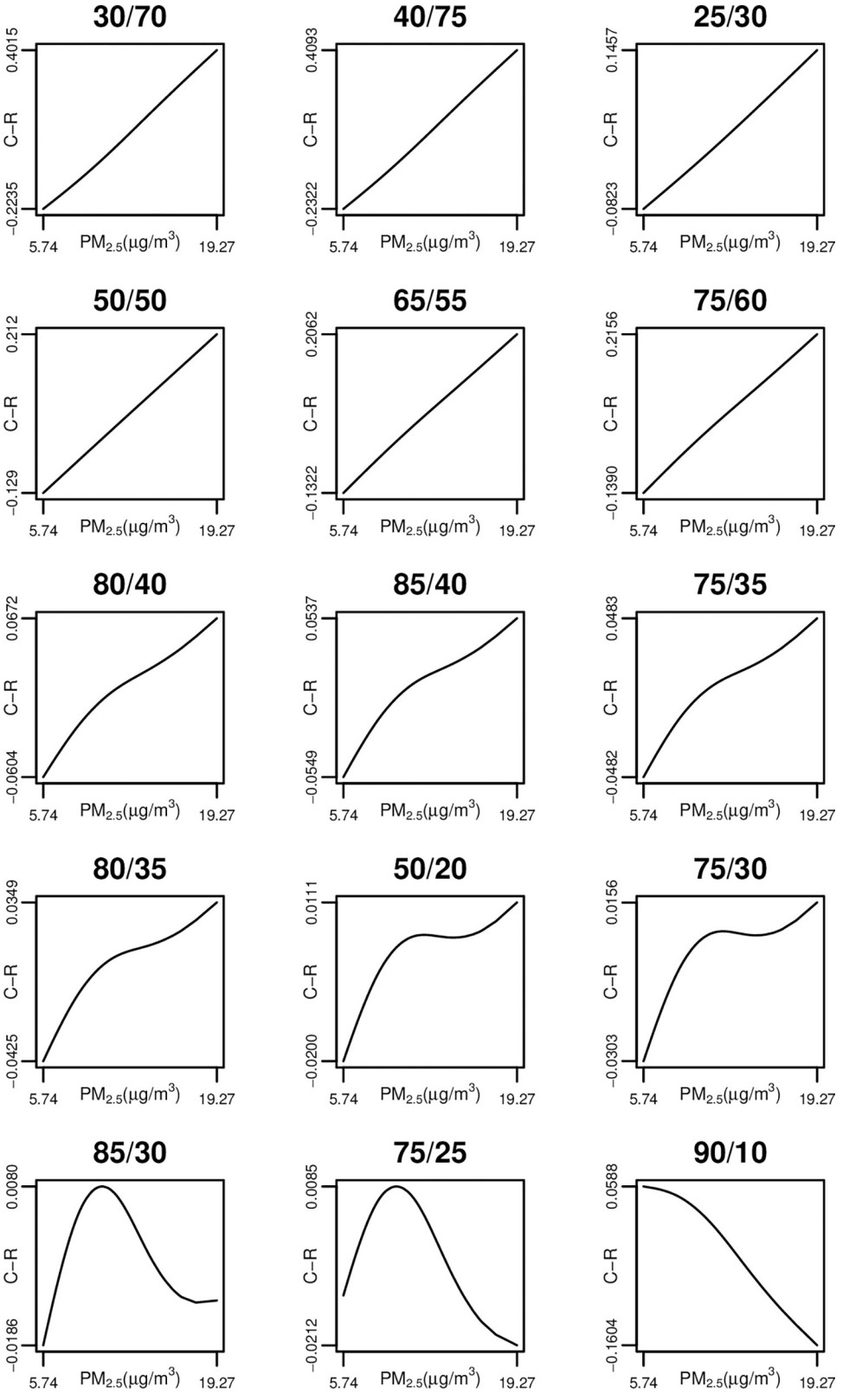

**Fig 4. Examples of C-R functions (PM$_{2.5}$ composition simulations, % toxic (City 1/City 100).**

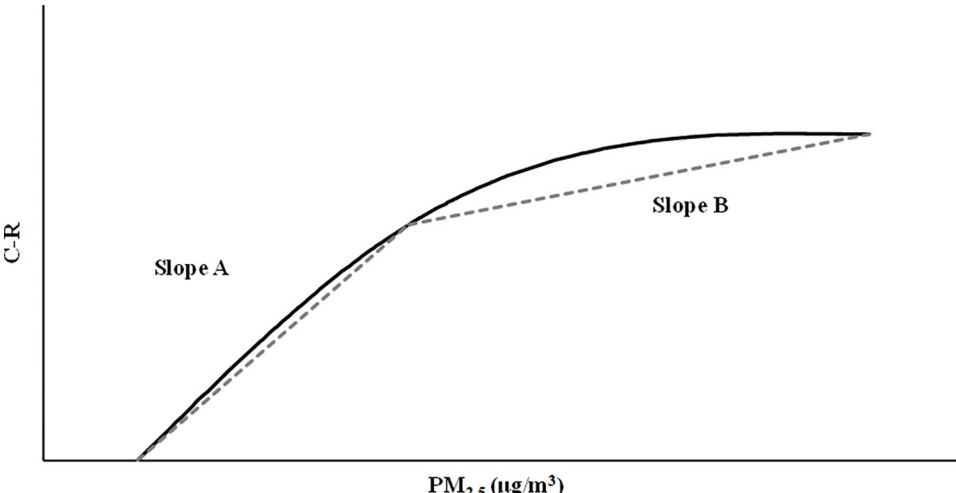

**Fig 5. The relative slope test for supralinearity.**

We also test for supralinearity by considering the relative improvement in model fit of a Cox PH model with a natural cubic spline as compared to a Cox PH model that assumes linearity. Previous studies have used this approach to assess supralinearity [10,12]. While a Cox PH model with a spline may produce an estimated hazard ratio that appears to be non-linear, it may not offer a statistically significantly better fit to the data than a Cox PH model that assumes linearity. We assess relative model fit by subtracting the Akaike Information Criterion (AIC) of the Cox PH model with a spline from the AIC of the linear Cox PH model. If the difference in the AICs is greater than two, we reject linearity and judge the Cox PH model with a spline to be a better fit to the data, thus accepting the apparently supralinear C-R function as a better fit to the data than the linear C-R function [21].

## 3. Results

### 3.1. Detecting true supralinearity

We begin by confirming that the methods we use to assess the shape of the C-R function do in fact detect true supralinearity when it exists. The results of these simulations are presented in Table 1. Each row of results in Table 1 presents one set of 10 simulations based on C-R functions with varying degrees of supralinearity, as shown in Fig 3. Each entry in Table 1 is the number of simulations out of 10 in which the named test (the slope ratio test or the difference in AIC test) detected a supralinear C-R function.

Examination of Table 1 reveals that, with one exception, both the relative slope ratio test and the AIC test indicate supralinear C-R functions in all 10 of the simulations when the C-R function is in fact supralinear. Further, these tests do not indicate supralinearity in any of the

**Table 1. Detecting true supralinearity with splines in a cox proportional hazards model.**

| Scale | Count Supralinear (Slope Ratio Test) | Count Supralinear (Difference in AIC Test) |
|---|---|---|
| 1 | 0 | 10 |
| 3 | 10 | 10 |
| 5 | 10 | 10 |
| Linear | 0 | 0 |

10 simulations when the true C-R function is linear. The one exception to the successful detection of supralinearity occurs in the most extreme case (scale = 1), in which random variation in mortality across the cities with the highest relative risks causes the estimated splines to bend downward at the highest PM$_{2.5}$ concentrations. In this case the estimated C-R function resembles an inverted "U" shape rather than the supralinear C-R functions that are more commonly reported in the literature.

## 3.2. PM$_{2.5}$ composition

The simulation results that examine the effect of PM$_{2.5}$ composition indicate that if the fraction of the mass that is toxic is higher in areas with lower total PM$_{2.5}$ mass, the estimated C-R function based on total PM$_{2.5}$ mass may appear to be supralinear. The results of our simulations examining this hypothesis are presented in Table 2. Each row of results in Table 2 presents one of the 30 sets of simulations we undertook. Each entry in Table 2 is the number of simulations out of 10 in which the named test (the slope ratio test or the difference in AIC test) detected a supralinear C-R function.

**Table 2. PM$_{2.5}$ composition effects on supralinearity.**

| Relative Toxicity (City 1/City 100) | % Toxic City 1 | % Toxic City 100 | Count Supralinear (Slope Ratio Test) | Count Supralinear (Difference in AIC Test) |
|---|---|---|---|---|
| 0.43 | 30 | 70 | 0 | 10 |
| 0.53 | 40 | 75 | 0 | 10 |
| 0.83 | 25 | 30 | 0 | 0 |
| 1.00 | 50 | 50 | 0 | 0 |
| 1.00 | 100 | 100 | 0 | 0 |
| 1.18 | 65 | 55 | 1 | 2 |
| 1.25 | 75 | 60 | 1 | 5 |
| 1.36 | 75 | 55 | 4 | 6 |
| 1.38 | 55 | 40 | 4 | 4 |
| 1.46 | 95 | 65 | 7 | 10 |
| 1.50 | 75 | 50 | 9 | 10 |
| 1.50 | 90 | 60 | 10 | 10 |
| 1.63 | 65 | 40 | 10 | 10 |
| 1.67 | 25 | 15 | 7 | 2 |
| 1.67 | 75 | 45 | 10 | 10 |
| 1.75 | 70 | 40 | 10 | 10 |
| 1.88 | 75 | 40 | 10 | 10 |
| 2.00 | 70 | 35 | 10 | 10 |
| 2.00 | 80 | 40 | 10 | 10 |
| 2.13 | 85 | 40 | 10 | 10 |
| 2.14 | 75 | 35 | 10 | 10 |
| 2.29 | 80 | 35 | 10 | 10 |
| 2.33 | 70 | 30 | 10 | 10 |
| 2.50 | 50 | 20 | 10 | 10 |
| 2.50 | 75 | 30 | 10 | 10 |
| 2.71 | 95 | 35 | 0 | 10 |
| 2.83 | 85 | 30 | 0 | 10 |
| 3.00 | 30 | 10 | 0 | 7 |
| 3.00 | 75 | 25 | 0 | 10 |
| 9.00 | 90 | 10 | 0 | 10 |

Table 2 is organized by the relative toxicity of the total PM$_{2.5}$ mass in City 1 (the city with the lowest PM concentration) to City 100 (the city with the highest PM concentration). The entries for City 1 and City 100 indicate the amount of total PM$_{2.5}$ mass that is toxic in each of those cities. For example, the simulations presented in the first row of Table 2 are based on a scenario where 30 percent of total PM$_{2.5}$ mass in City 1 was toxic, while 70 percent of total PM$_{2.5}$ mass in City 100 was toxic (this corresponds to a concentration of toxic PM$_{2.5}$ of 1.72 μg/m$^3$ in City 1, and 13.49 μg/m$^3$ in City 100). The percentages of toxic PM$_{2.5}$ for the remaining cities were calculated through linear interpolation, as described above. In all cases the hazard ratio associated with toxic PM$_{2.5}$ was 1.05.

For each of the 30 scenarios presented in Table 2, we undertook 10 simulations. In each row we report the number of simulations that indicated apparent supralinearity by the relative slope test and by the AIC test.

Examination of Table 2 reveals that apparent supralinearity is most common when the relative toxicity of the total PM$_{2.5}$ mass in the city with the lowest PM$_{2.5}$ concentration to the city with the highest PM$_{2.5}$ concentration ranges between 1.5 and 2.5. For nearly all of the scenarios in this range, both the relative slope test and the AIC test indicate apparent supralinearity for all 10 simulations based on the scenario. The one exception in this range occurs when the toxic percent of the total PM$_{2.5}$ mass is very low for all cities (25 percent in City 1 and 15 percent in City 100).

Below this range, apparent supralinearity becomes much less common. When the relative toxicity of City 1 to City 100 equals 1, the apparent C-R function will be linear, and when this relative toxicity falls below 1 we generally estimate sublinear C-R functions. Similarly, supralinearity becomes much less common when the relative toxicity of the total PM$_{2.5}$ mass in the city with the lowest PM concentration to the city with the highest PM concentration is higher than 2.5. In these cases, the absolute amount of toxic PM approaches equality across cities, leading to flat but near-linear C-R functions that are not supralinear by the relative slope test (although they are still distinguishable from a linear C-R function through the AIC test).

## 3.3. Risk factor correlated with PM$_{2.5}$

The results for the simulations based on PM$_{2.5}$ exposure serving as a proxy for some other risk factor are similar to those from the PM$_{2.5}$ composition simulations described above. The results of our simulations examining this hypothesis are presented in Table 3. Each row of Table 3 presents one of the 44 sets of simulations we undertook. For each of the 44 scenarios presented in Table 3, we undertook 10 simulations, with the number of simulations that indicated apparent supralinearity by the relative slope test and by the AIC test reported in each row.

Table 3 is organized by the amount of the risk factor relative to the PM$_{2.5}$ concentrations in City 1 as compared to City 100. For example, the simulations presented in the first row of Table 3 are based on a scenario where the concentration of the risk factor is equivalent to 30 percent of the PM$_{2.5}$ concentration in City 1, and 70 percent of the PM$_{2.5}$ concentration in City 100. The percentages of the alternate risk factor for the remaining cities were calculated through linear interpolation, as described above. In all cases the hazard ratio associated with this alternate risk factor was 1.05.

Examination of Table 3 reveals that, as with the simulations based on PM$_{2.5}$ composition, apparent supralinearity is most common when the amount of the risk factor relative to the PM$_{2.5}$ concentration in the city with the lowest PM concentration to the city with the highest PM concentration ranges between 1.5 and 2.5. For nearly all scenarios in this range, both the relative slope test and the AIC test indicated apparent supralinearity for all 10 simulations.

**Table 3. Non-PM$_{2.5}$ risk factor effects on supralinearity.**

| Relative Toxicity (City 1/City 100) | % of PM$_{2.5}$ City 1 | % of PM$_{2.5}$ City 100 | Count Supralinear (Slope Ratio Test) | Count Supralinear (Difference in AIC Test) |
|---|---|---|---|---|
| 0.43 | 30 | 70 | 0 | 10 |
| 0.53 | 40 | 75 | 0 | 10 |
| 0.83 | 25 | 30 | 0 | 0 |
| 1.00 | 50 | 50 | 0 | 0 |
| 1.00 | 100 | 100 | 0 | 0 |
| 1.00 | 150 | 150 | 0 | 1 |
| 1.18 | 65 | 55 | 1 | 2 |
| 1.25 | 75 | 60 | 1 | 5 |
| 1.25 | 125 | 100 | 0 | 9 |
| 1.25 | 150 | 120 | 0 | 10 |
| 1.36 | 75 | 55 | 4 | 6 |
| 1.36 | 150 | 110 | 0 | 10 |
| 1.38 | 55 | 40 | 4 | 4 |
| 1.46 | 95 | 65 | 7 | 10 |
| 1.50 | 75 | 50 | 9 | 10 |
| 1.50 | 90 | 60 | 10 | 10 |
| 1.50 | 150 | 100 | 10 | 10 |
| 1.56 | 125 | 80 | 10 | 10 |
| 1.63 | 65 | 40 | 10 | 10 |
| 1.67 | 25 | 15 | 7 | 2 |
| 1.67 | 75 | 45 | 10 | 10 |
| 1.67 | 125 | 75 | 10 | 10 |
| 1.67 | 150 | 90 | 10 | 10 |
| 1.75 | 70 | 40 | 10 | 10 |
| 1.79 | 125 | 70 | 10 | 10 |
| 1.88 | 75 | 40 | 10 | 10 |
| 1.88 | 150 | 80 | 10 | 10 |
| 2.00 | 70 | 35 | 10 | 10 |
| 2.00 | 80 | 40 | 10 | 10 |
| 2.13 | 85 | 40 | 10 | 10 |
| 2.14 | 75 | 35 | 10 | 10 |
| 2.29 | 80 | 35 | 10 | 10 |
| 2.33 | 70 | 30 | 10 | 10 |
| 2.50 | 50 | 20 | 10 | 10 |
| 2.50 | 75 | 30 | 10 | 10 |
| 2.71 | 95 | 35 | 0 | 10 |
| 2.83 | 85 | 30 | 0 | 10 |
| 3.00 | 30 | 10 | 0 | 7 |
| 3.00 | 75 | 25 | 0 | 10 |
| 3.00 | 150 | 50 | 0 | 10 |
| 3.13 | 125 | 40 | 0 | 10 |
| 6.00 | 150 | 25 | 0 | 10 |
| 6.25 | 125 | 20 | 0 | 10 |
| 9.00 | 90 | 10 | 0 | 10 |

This holds true even though additional scenarios are considered in this set of simulations under which the relative amount of the alternate risk factor could be larger than 100 percent. The one exception in this range occurs when the relative amount of the risk factor very low for all cities. Above and below the range of 1.5 to 2.5, apparent supralinearity becomes much less common, for the reasons described above.

### 3.4. Measurement error

The results examining the effect of measurement error on bias in shape estimates indicate that if exposure misclassification is disproportionately large in areas with higher PM$_{2.5}$ concentrations, the estimated C-R function may falsely appear to be supralinear. The results of the relative slope ratio test for this positive correlation case are presented in Fig 6 for draws of the observed PM$_{2.5}$ values truncated at ± 5 μg/m$^3$, for 3 of the hazard ratios we considered (1.005, 1.01, and 1.03). Each dot in this figure represents the result from one simulation (100 simulations were undertaken for each hazard ratio and type of measurement error). Recall that these tests are assessing apparent supralinearity relative to the "true" C-R function without measurement error, and this "true" C-R function may not be perfectly linear. The vertical axis in Fig 6 indicates the percentage change in relative slope caused by measurement error.

Examination of Fig 6 reveals that for relatively small amounts of measurement error (5 percent of the true concentration of PM$_{2.5}$), a 20 percent increase in the relative slope ratio occurs less than 10 percent of the time. A shift in the direction of false indication of supralinearity is much more common when measurement error is 10 percent of the true concentration of PM$_{2.5}$, with an increase in the supralinearity in the C-R function occurring in about 40 percent of our simulations. However, at high levels of measurement error (20 percent of the true concentration of PM$_{2.5}$), shifts in both the supralinear and sublinear direction become possible, as the noise in the measurement error overwhelms the shape of the "true" underlying C-R function (which is linear). This is especially true when the association between true PM$_{2.5}$ and mortality is weak (hazard ratio = 1.005).

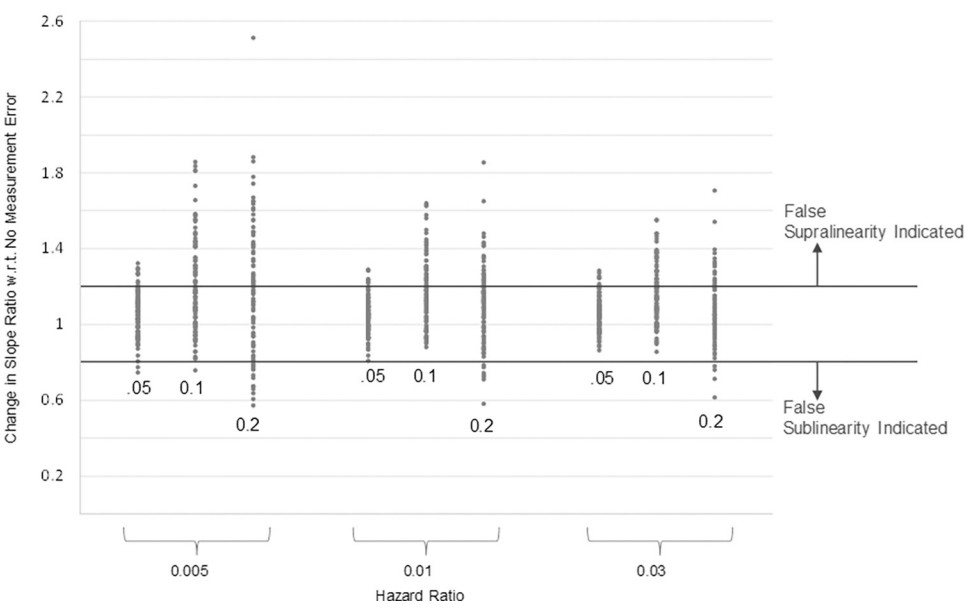

**Fig 6. Effect of measurement error on change in slope ratio compared to no measurement error, positive correlation case.**

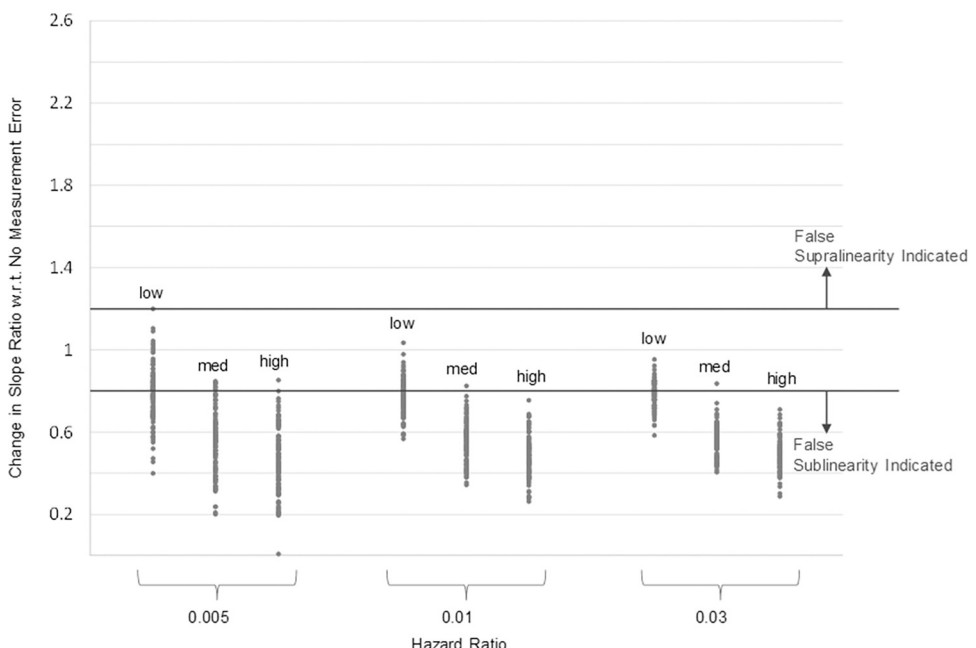

**Fig 7. Effect of measurement error on change in slope ratio compared to no measurement error, negative correlation case.**

The simulation results for cases where exposure misclassification is disproportionately small in areas with higher PM$_{2.5}$ concentrations are presented in Fig 7, which is organized in the same way as Fig 6. These simulation results reveal that in this negative correlation case, increases in the apparent supralinearity of the C-R functions almost never occurs, and even then, only happens in cases where random variation in mortality is most influential (hazard ratio = 1.005). Instead, more than half of the estimated C-R functions are shifted in a sublinear direction for the lowest level of measurement error, and nearly all the estimated C-R functions are shifted in a sublinear direction at the medium and high levels of measurement error.

For both the positive and negative correlation cases, we also undertook simulations with draws of the observed PM$_{2.5}$ values from normal distributions truncated at $\pm$ 4 µg/m$^3$ and $\pm$ 6 µg/m$^3$, and for a hazard ratio of 1.02. These results are presented in the SI as 1 and S2 Tables.

## 4. Discussion

Several previous studies have produced estimates of apparently supralinear C-R functions for the relationship between chronic mortality risk and PM$_{2.5}$ [3,8,10–14]. However, these studies treat these C-R function estimates as causal relationships, and do not (indeed, cannot) examine alternative explanations for the apparent supralinearity. In this study, we considered several alternative explanations for apparent supralinearity in estimates of C-R functions for PM$_{2.5}$. To the best of our knowledge, this is the first simulation-based study to examine alternative explanations for apparent supralinearity in statistical estimates of C-R functions.

In one set of simulations, we found that the C-R relationship between PM$_{2.5}$ and mortality could falsely appear to be supralinear when the fraction of the mass that is toxic is higher in areas with lower total PM$_{2.5}$ mass as compared to areas with higher total PM$_{2.5}$ mass. Previous research has shown that differences in PM$_{2.5}$ composition may affect the association between PM$_{2.5}$ and mortality [22]. This scenario could plausibly apply to the U.S., as both the total mass

and the composition of ambient PM$_{2.5}$ in the U.S. varies across regions and seasons. For example, nitrates are a larger fraction of total PM$_{2.5}$ mass in the Western U.S., while sulfates are a larger fraction of total PM$_{2.5}$ mass in the Eastern U.S. [23]. These differences in composition across regions could lead to scenarios such as those we describe in our simulations. The relationship between total PM$_{2.5}$ mass, PM$_{2.5}$ composition, and mortality may not be immediately obvious, especially if the constituent components in PM$_{2.5}$ interact in some way. Further research is needed on the variation in PM$_{2.5}$ composition across communities, and the mortality risks associated with specific components of PM$_{2.5}$.

In another set of simulations, we found that the C-R relationship between PM$_{2.5}$ and mortality also could falsely appear to be supralinear when PM$_{2.5}$ exposure serves as a proxy for some other risk factor that is disproportionately large in areas with higher PM$_{2.5}$ concentrations. Again, this scenario could plausibly apply to the U.S. Lower socioeconomic status (SES) is linked to both cardiovascular risk factors (such as obesity, physical inactivity, and smoking) and higher exposure to ambient PM$_{2.5}$ [24,25]. Previous research has found higher mortality risks related to exposure to PM$_{2.5}$ for self-identified racial minorities and people with low income [12], which could be explained at least in part by the effect of other variables associated with SES. Further research is needed on the influence of SES-related and other possible confounding factors on the estimated C-R relationship between PM$_{2.5}$ and mortality.

In our third set of simulations, we found that the C-R relationship between PM$_{2.5}$ and mortality could falsely appear to be supralinear when exposure misclassification (measurement error) is disproportionately large in areas with higher PM$_{2.5}$ concentrations. It is well-known that the "classical-type" measurement error we examined here is present to some degree in measures of ambient PM$_{2.5}$, and this in turn can influence the estimated shape of the C-R function [15,16,20]. Less is known about the relationship between exposure misclassification and ambient PM$_{2.5}$ concentrations. However, recent research suggests that the differences between outdoor air quality monitors (the typical source of data for ambient PM$_{2.5}$ concentrations in cohort studies) and personal measures of PM$_{2.5}$ exposure tend to be larger when the level of ambient PM$_{2.5}$ as measured by the monitor is higher [26]. This is consistent with the type of measurement error we considered in our simulation scenarios, but further research on the relationship between exposure misclassification and ambient PM$_{2.5}$ concentrations is needed, especially given the increasing use of reconstructed PM$_{2.5}$ exposure measures, such as those based on remote sensing in areas with few air quality monitors [27]. We also note that some types of systematic measurement error (such as systematically overestimating PM$_{2.5}$ exposure by a constant percentage) could also induce apparent supralinearity—this is an area for future research.

While we have considered three alternative explanations for apparent supralinearity in C-R functions here, there are of course other possible explanations. For example, another possible source of apparent supralinearity is simple random variability in mortality. To test the effect of simple random variability in mortality within a cohort on the appearance of apparent supralinearity, for each cohort we examined the shape of the estimated C-R function with only 1 year of follow-up, rather than the 20 years of follow-up used in our tests above–cohorts with only 1 year of follow-up will have more random variability than cohorts with 20 years of follow-up. We undertook this test for all the simulations described above except for the measurement error simulations. The results of the simulations based on cohorts with one 1 year of follow up reveal that the ability to detect true supralinearity declines, and the chance of falsely estimating supralinearity increases, as the amount of random variability in the cohort increases (these results are presented in the SI as S3–S5 Tables). Adding additional demographic groups or non-PM city-specific mortality effects to the simulations also increased the amount of random variability in the cohort, and sometimes led to false estimates of supralinearity. Further

research is needed on how random variability in mortality affects the ability to estimate the true C-R relationship, especially since apparent supralinearity has been observed in some short-term mortality studies [28].

Overall, our results show that a linear C-R function for PM$_{2.5}$ can falsely appear to be supralinear for a variety of reasons, such as spatial variation in the composition of total PM$_{2.5}$ mass, the presence of confounders that are correlated with PM$_{2.5}$ exposure, and some types of measurement error in estimates of PM$_{2.5}$ exposure. Determining whether the C-R function for PM$_{2.5}$ is truly supralinear has important implications both for setting the NAAQS for particulate matter and for setting policies related to the attainment of air quality standards. Alternative explanations for apparent supralinearity should be more carefully considered and quantified in future research.

## Supporting information

**S1 Table. Effect of measurement error on change in slope ratio compared to no measurement error, positive correlation case.**
(DOCX)

**S2 Table. Effect of measurement error on change in slope ratio compared to no measurement error, negative correlation case.**
(DOCX)

**S3 Table. Detecting true supralinearity with splines in a cox proportional hazards model, 1 year follow-up.**
(DOCX)

**S4 Table. PM$_{2.5}$ composition effects on supralinearity, 1 year follow-up.**
(DOCX)

**S5 Table. Non-PM$_{2.5}$ risk factor effects on supralinearity, 1 year follow-up.**
(DOCX)

## Acknowledgments

We thank Annette Rohr for comments on an earlier version of this manuscript.

## Author Contributions

**Conceptualization:** Garrett Glasgow, Anne E. Smith.

**Methodology:** Garrett Glasgow, Bharat Ramkrishnan, Anne E. Smith.

**Validation:** Garrett Glasgow, Bharat Ramkrishnan, Anne E. Smith.

**Visualization:** Garrett Glasgow, Bharat Ramkrishnan, Anne E. Smith.

**Writing – original draft:** Garrett Glasgow, Bharat Ramkrishnan, Anne E. Smith.

**Writing – review & editing:** Garrett Glasgow, Bharat Ramkrishnan, Anne E. Smith.

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
