## [Decision Letter · Decision Letter 0]

12 Mar 2024

PONE-D-23-43296Model Misspecification, Measurement Error, and Apparent Supralinearity in the Concentration-Response Relationship between PM_2.5_ and MortalityPLOS ONE

Dear Dr. Glasgow,

Thank you for submitting your manuscript to PLOS ONE. After careful consideration, we feel that it has merit but does not fully meet PLOS ONE’s publication criteria as it currently stands. Therefore, we invite you to submit a revised version of the manuscript that addresses the points raised during the review process.

We look forward to receiving your revised manuscript.

Kind regards,

Worradorn Phairuang, Ph.D.

Academic Editor

PLOS ONE

Journal Requirements:

"This work was funded by the Electric Research Power Institute (EPRI)."

"GG, BR, and AS were funded by the Electric Power Research Institute (EPRI).

URL: https://www.epri.com/

The funders had no role in study design, data collection and analysis, decision to

publish, or preparation of the manuscript."

**Additional Editor Comments:**

ACADEMIC EDITOR: Major revision.

Reviewers' comments:

Reviewer's Responses to Questions

**Comments to the Author**

1. Is the manuscript technically sound, and do the data support the conclusions?

Reviewer #1: Yes

Reviewer #2: Partly

2. Has the statistical analysis been performed appropriately and rigorously? 

Reviewer #1: Yes

Reviewer #2: Yes

3. Have the authors made all data underlying the findings in their manuscript fully available?

Reviewer #1: No

Reviewer #2: No

4. Is the manuscript presented in an intelligible fashion and written in standard English?

Reviewer #1: Yes

Reviewer #2: Yes

5. Review Comments to the Author

Reviewer #1: This is a simulation study which examine alternative hypotheses (to the one of actual causation) for the apparent supralinearity of the concentration response (C-R) relation between population PM2.5 exposure and mortality. According to the manuscript, three alternative explanations for apparent supralinearity in a C-R function are tested.

The first alternative explanation is whether spatial variation in the composition of total PM2.5 mass can cause the false appearance of supralinearity in statistical estimates. Under this scenario, total PM2.5 mass is composed of both toxic and non-toxic constituents, with the fraction of the mass that is toxic affecting mortality through a linear C-R function. Even though the true C-R function for toxic PM2.5 is linear, if the fraction of the mass that is toxic is higher in areas with lower total PM2.5 mass, the estimated C-R function based on total PM2.5 mass may appear to be supralinear.

The second alternative explanation we test is whether apparent supralinearity can be arise in cases where PM2.5 exposure has been serving as a proxy for some other risk factor. Under this scenario, if the alternative risk factor is disproportionately large in areas with higher PM2.5 concentrations, the estimated C-R function for PM2.5 may appear to be supralinear, even if the true C-R function for the alternative risk factor is linear.

The third alternative explanation we test examines whether exposure misclassification (classical measurement error) related to measured PM2.5 concentrations can result in apparent supralinearity in statistical estimates. Two patterns of measurement error are examined, one under which exposure misclassification is disproportionately large in areas with higher PM2.5 concentrations, and one under which exposure misclassification is disproportionately small in areas with higher PM2.5 concentrations. If the measurement error varies across the range of measured PM2.5 concentrations, this could attenuate some portions of the C-R function more than others, leading to apparent supralinearity in the estimation of the C-R function.

The three alternative explanations are all reasonable and worth investigating (although they are not the only ones which could explain the apparent supralinearity of the relation), and the simulation study shows that all the scenarios may explain why supralinearity is apparent.

To build their simulated cohorts, the authors say that "The baseline mortality rate for the simulated individuals in our cohort was calculated using cohort life tables compiled by the US Social Security Administration (Bell and Miller 2005, Table 7). These life tables give the probability of mortality at each age based on birth year and sex, with birth year reported in 10-year increments from 1900 to 2100". However, total mortality includes natural mortality and mortality due to other causes (car accidents, natural disasters, homicides and so on) which may not be influenced by PM2.5 exposure: thus, if they used total mortality, they should state this clearly and acknowledge this fact as a limitation of the study (or, better, run again the simulation using natural mortality rates).

Another issue is that the simulated cohorts are composed using only males data: as, apparently, mortality assumed to be related to air pollution levels has been shown, in epidemiological studies, to be possibly different between males and females, the authors should discuss this issue.

Reviewer #2: The manuscript "Model Misspecification, Measurement Error, and Apparent Supralinearity in the

Concentration-Response Relationship between PM2.5 and Mortality" presents a simulation study of cohort analyses, providing illustrations of sources of potential supralineairty in the exposure-response function of PM2.5 on mortality.

The exercise is very nice and I think it is a good idea to perform this kind of simulations as it can help understanding where models can be misleading. So I think the objective of the paper is very relevant. I have nonetheless some comments that I think should be addressed in the study.

# Major

1. A lot of the results rely on the test of supralinearity and I am not sure it has been explained enough.

1.1. In particular, the thresholds for "significance" (1.2) is not really justified and does not seem to rely on any asymptotic result. It would be useful to have an idea of its power in different situations.

1.2. An alternative to this test could a simpler Wald test based on second derivative of the C-R function. Estimating the C-R with B-splines, one can show that a concave function would mean negative second differences between the B-Spline coefficients (say d2 beta, see e.g. https://doi.org/10.1007/s11222-013-9448-7). From there a Wald test testing H0: d2 beta = 0 (linearity) vs H1 d2 beta < 0 (supralinearity) can do the trick.

1.3 In addition to the test, I think it would useful to display the estimate C-R curves for many scenarios. It would provide an idea of to which extent the scenario creates supralinearity where there is none.

2. Performing the simulations is interesting to understand how models behave in various situations, but I feel the models applied here do not really reflect the current practice in long-term PM2.5 studies. My understanding from the study is that Cox models are applied with no confounders or stratification. But in practice, many studies will stratify by location (assessment center, city, ...) or include some proxy as a confounder in the model. In this case, would the apparent supralinearity persist nonetheless?

3. About the PM2.5 composition scenarios

3.1. I would be curious to see some scenario where the differences in PM2.5 toxicity is independent from the total PM2.5 mass. The idea behind the PM2.5 composition studies is that, for fixed PM2.5 levels there are probably different vulnerability because of the relative toxicity of the local mix.

3.2. Considering the "proportion of toxicity" is fine as a simplified model, but it could be acknowledged that potentially all PM2.5 component are toxic (even the natural ones such as desert dust), they just are toxic to different degrees.

4. About the measurement error scenario:

4.1. You could argue that this one is becoming important with the advent of reconstructed exposure datasets (e.g. https://doi.org/10.3390/rs12223803), that probably display higher error than a monitoring station.

4.2. I would expect that a common error would actually be a systematic bias. Would the results hold in such a case?

# Minor

- The paper focuses on long-term studies, but note that short-term studies also suggest potential nonlinearities (e.g. https://doi.org/10.1056/NEJMoa1817364).

- p.7 line 1: a simpler way to explain this is sampling from a Bernoulli with p being the probability of mortality

- The model linking exposure to hazard ratio is not clear as the exposure appears nowhere in the equations.

- I am not sure I understand what the authors mean when they mention that the true C-R is not perfectly linear (p. 13 l. 4). Aren't we trying to estimate the hazard ratio h, which represents a linear relationship in the simulations?

6. PLOS authors have the option to publish the peer review history of their article (what does this mean?). If published, this will include your full peer review and any attached files.

Reviewer #1: No

Reviewer #2: No

---

## [Author Response · Author response to Decision Letter 0]

16 Apr 2024

We have addressed all editor and reviewer comments in our cover letter and in the Response to Reviewers.

---

## [Decision Letter · Decision Letter 1]

30 Apr 2024

Model Misspecification, Measurement Error, and Apparent Supralinearity in the Concentration-Response Relationship between PM_2.5_ and Mortality

PONE-D-23-43296R1

Dear Dr. Glasgow,

We’re pleased to inform you that your manuscript has been judged scientifically suitable for publication and will be formally accepted for publication once it meets all outstanding technical requirements.

Kind regards,

Worradorn Phairuang, Ph.D.

Academic Editor

PLOS ONE

Additional Editor Comments (optional):

Reviewers' comments:

Reviewer's Responses to Questions

**Comments to the Author**

1. If the authors have adequately addressed your comments raised in a previous round of review and you feel that this manuscript is now acceptable for publication, you may indicate that here to bypass the “Comments to the Author” section, enter your conflict of interest statement in the “Confidential to Editor” section, and submit your "Accept" recommendation.

Reviewer #1: (No Response)

2. Is the manuscript technically sound, and do the data support the conclusions?

Reviewer #1: Yes

3. Has the statistical analysis been performed appropriately and rigorously? 

Reviewer #1: Yes

4. Have the authors made all data underlying the findings in their manuscript fully available?

Reviewer #1: Yes

5. Is the manuscript presented in an intelligible fashion and written in standard English?

Reviewer #1: Yes

6. Review Comments to the Author

Reviewer #1: (No Response)

7. PLOS authors have the option to publish the peer review history of their article (what does this mean?). If published, this will include your full peer review and any attached files.

Reviewer #1: No

---

## [Editor Report · Acceptance letter]

4 May 2024

PONE-D-23-43296R1 

PLOS ONE

Dear Dr. Glasgow, 

I'm pleased to inform you that your manuscript has been deemed suitable for publication in PLOS ONE. Congratulations! Your manuscript is now being handed over to our production team.

Kind regards, 

on behalf of

Assistant Professor Worradorn Phairuang 

Academic Editor

PLOS ONE